# A Resilience Related Glial-Neurovascular Network Is Transcriptionally Activated after Chronic Social Defeat in Male Mice

**DOI:** 10.3390/cells11213405

**Published:** 2022-10-27

**Authors:** Constance Vennin, Charlotte Hewel, Hristo Todorov, Marlon Wendelmuth, Konstantin Radyushkin, André Heimbach, Illia Horenko, Sarah Ayash, Marianne B. Müller, Susann Schweiger, Susanne Gerber, Beat Lutz

**Affiliations:** 1Leibniz Institute for Resilience Research, 55122 Mainz, Germany; 2Institute of Physiological Chemistry, University Medical Center of the Johannes Gutenberg University Mainz, 55128 Mainz, Germany; 3Institute of Human Genetics, University Medical Center of the Johannes Gutenberg University Mainz, 55131 Mainz, Germany; 4Institute of Human Genetics, University of Bonn and University Hospital Bonn, 53127 Bonn, Germany; 5Faculty of Informatics, Università della Svizzera Italiana, 6900 Lugano, Switzerland; 6Department of Psychiatry, Psychotherapy, and Focus Program Translational Neurosciences, Translational Psychiatry, University Medical Center of the Johannes Gutenberg University Mainz, 55128 Mainz, Germany

**Keywords:** single-cell RNA-seq, stress resilience, hippocampus, neuroimmune pathways, cell-cell interaction, glial cells, neurovascular system

## Abstract

Upon chronic stress, a fraction of individuals shows stress resilience, which can prevent long-term mental dysfunction. The underlying molecular mechanisms are complex and have not yet been fully understood. In this study, we performed a data-driven behavioural stratification together with single-cell transcriptomics of the hippocampus in a mouse model of chronic social defeat stress. Our work revealed that in a sub-group exhibiting molecular responses upon chronic stress, the dorsal hippocampus is particularly involved in neuroimmune responses, angiogenesis, myelination, and neurogenesis, thereby enabling brain restoration and homeostasis after chronic stress. Based on these molecular insights, we applied rapamycin after the stress as a proof-of-concept pharmacological intervention and were able to substantially increase stress resilience. Our findings serve as a data resource and can open new avenues for further understanding of molecular processes underlying stress response and for targeted interventions supporting resilience.

## 1. Introduction

Chronic stress challenges human health. Interestingly, most individuals considered as resilient are resistant to developing stress-induced mental dysfunction [1,2,3,4]. However, a small proportion of individuals, designated as susceptible, develop stress-related disorders such as anxiety and depression. Mechanistic understanding of why some individuals are resilient and others are not is of high medical interest nowadays. Indeed, after two years of COVID-19, it has been reported that the number of major depressive and anxiety disorder cases has increased by 27.6% and 25.6%, respectively, in 204 countries and territories [5]. These observations show the urgency of identifying the molecular mechanisms of stress resistance and developing new appropriate treatments.

Stress resilience is considered as an active and adaptive process during adversity, trauma, or threats, preventing major depressive disorders [6,7,8,9]. The underlying mechanisms that are decisive for resilient or susceptible behaviour have not yet been understood in detail. Elucidating the molecular basis of the various responses to chronic stress is therefore crucial and paves the way towards targeted interventions to strengthen resilience and to prevent mental dysfunction [6,7,8,9]. An essential step towards the mechanistic understanding of resilience was the implementation of the chronic social defeat stress (CSDS) mouse model [10]. The original paradigm, consisting of CSDS followed by the social interaction (SI) test, identified two groups of defeated mice, respectively, the resilient and the stress-impaired susceptible mice [10,11,12,13]. Using this model, several post-CSDS bulk transcriptome analyses on different brain regions, including the prefrontal cortex (PFC), nucleus accumbens (NAc), and ventral hippocampus, identified neuronally expressed genes that appear to be involved in resilience behaviour [10,11,12]. However, in rodents and in humans, 50% of the brain cells are glial cells, i.e., astrocytes, oligodendrocytes and microglial cells, highlighting the importance of these cells for brain function [14]. Indeed, in many brain disorders, alterations of the functions of non-neuronal cells can lead to neuronal dysfunction and to disorders [15,16]. Recently, it was shown that CSDS affects the integrity of the blood-brain barrier (BBB) and that non-neuronal cells, i.e., oligodendrocytes and astrocytes, are involved in the protection against chronic stress-related pathologies [17,18,19,20]. 

In the commonly used model of CSDS, both the susceptible and resilient groups of mice show a large variability in terms of the distribution of the SI indices after CSDS [10], suggesting that additional sub-groups might exist within these two populations. Recently, the mode of outcome measures after CSDS has been refined by replacing the commonly used SI test by a modified social interaction (MSI) test, allowing us to unravel an additional, hitherto hidden subgroup of defeated mice, exhibiting phenotypic characteristics of resilience [21,22]. Furthermore, a recent longitudinal study in mice indicated that the behavioural outcome after CSDS features dynamic processes. Mice initially classified as resilient or susceptible, respectively, could change their behavioural outcome later in the observation period [23], similar to reports for humans [24,25].

Here, we applied a refined behaviour model that not only included the dichotomy of resilience versus susceptibility, but also considered the dynamics in behavioural response and the emergence of subgroups in the resilience outcome. We combined this model with in-depth single-cell RNA-sequencing (scRNAseq) of the dorsal and ventral hippocampus. Identifying transcriptional differences at the single-cell level within the different brain cell types allowed us to construct a high-resolution map of the regulatory networks that may direct the stress-induced cellular responses. Furthermore, the differentiation of hippocampal regions into dorsal and ventral parts allowed addressing the possible occurrence of region-specific molecular signatures upon CSDS. Clustering defeated animals into three distinct groups permitted us to investigate the molecular response in a subgroup of mice, defined as susceptible according to the criteria of Krishnan and colleagues [10], but which showed a less severely affected, intermediate behavioural phenotype. This sub-group exhibited transcriptional profiles characterised by intricate glial and neurovascular cell-cell signalling processes in the dorsal hippocampus. Furthermore, as a proof-of-concept pharmacological intervention, rapamycin applied after CSDS was able to promote stress resilience in mice. These data suggest a high degree of adaptability at the molecular level together with a less pronounced CSDS-induced social avoidance phenotype as an indicator of an ongoing active resilience process in the intermediate behavioural group. The major transcriptional changes observed in a subset of susceptible mice could serve as a resource for stress resilience research and the identification of new drugs potentially promoting resilience.

## 2. Materials and Methods

### 2.1. Animals 

C57BL/6J male mice of 8 weeks old (Janvier, France) were imported into the animal facility. Mice were single housed in a temperature and humidity-controlled room with 12 h light-dark cycle (lights on 7 a.m.–7 p.m.) and had access to food and water ad libitum. Mice were allowed to habituate for at least one week before CSDS. All experiments were carried out in accordance with the European Community’s council Directive of 22 September 2010 (2010/63EU) and were approved by the local animal care committee (Landesuntersuchungsamt Koblenz, Rhineland Palatinate, permit number 23 177-07/G 17-1-005, 23 177-07/G 17-1-049). 

### 2.2. Chronic Social Defeat Stress

To induce chronic social stress, we applied the procedure, which was developed based on an earlier published protocol [26,27]. Thus, every day for 10 days, mice from the defeated group were subjected to 3 social defeat sessions with a 15 min interval. During such sessions, a defeated mouse was introduced into a home cage of an older, larger, and more aggressive retired male breeder of the CD1 strain. After being physically defeated (attacked) for a total of 15 s, a mesh wall was introduced in the middle of the cage between the two mice, allowing sensory but not physical contact. After the last defeat session, the mesh was left until the next day, thus defeated and CD1 mice were housed separated in the same home cage. Prior to beginning the CSDS procedure, we selected CD1 males with attack latency less than 10 s. Mice from the non-defeated control group were housed in the same conditions (i.e., 2 mice per cage, separated by a mesh), and were handled daily throughout 10 days by being placed for about 30 s into an empty cage and then set back to their home cage (repeated three times per day). All cages were maintained in environmentally controlled cabinets (Uniprotect NG, Zoonlab GmbH, Castrop-Rauxel, Germany), which were located in different rooms for stressed (*n* = 58) and non-stressed (*n* = 12) groups. On the next day after the last defeat session, all animals were separated and housed individually before beginning of the behavioural testing.

### 2.3. Social Interaction Test

Social interaction test was performed as described by Golden and colleagues [26]. Thus, the mesh enclosure was placed touching the wall in the middle part of the side of a square-shaped (40 × 40 cm) open-field arena. On the day of testing, mice were introduced in the middle of the arena, first for 2.5 min of habituation with empty mesh enclosures, which was immediately followed by 2.5 min of testing with the novel (i.e., never used for defeat) CD1 mouse placed in the enclosures. Ethovision XT 15 (Noldus) system was used for video-tracking in order to measure the total presence time of the mouse’s mid-point in the interaction zone, defined as 8 cm from the enclosures’ boundaries. The social interaction index was calculated as followed: time spent exploring the mesh enclosure with the CD1 mouse during the testing phase/the time spent exploring the empty enclosure during the testing phase. 

### 2.4. Modified Social Interaction Test 

In order to investigate the specificity of CSDS-induced social avoidance towards CD1 males, we applied the modified social interaction (MSI) test as described by Ayash and colleagues [22]. Briefly, the test was performed in the arena, which is divided into three equal compartments, 20 × 40 cm each. Transparent plastic walls separated compartments with openings that allow mice to move in between. One mesh enclosure was placed at each of the outer compartments. The enclosure at one side contained a novel (not used for CSDS) CD1 retired breeder, whereas the other ones contained a 129/Sv mouse, matched in age to the CD1. Social targets’ locations were counterbalanced to deal with potential side preferences. At the day of testing, mice were introduced in the middle area of the arena twice, first for 6 min of habituation with empty mesh enclosures, which was immediately followed by 6 min of testing with the novel mice placed in the enclosures. Ethovision XT 15 (Noldus) system was used for video-tracking in order to measure the total time of presence of mouse nose in the interaction zone, defined as 3 cm from the enclosures’ boundaries. The SI index was calculated as followed: time spent exploring each mouse strain during the testing phase/average of time exploring the two empty mesh enclosures during the habituation phase. 

### 2.5. Drug Administration

Rapamycin (LC Laboratories, Woburn, MA, USA) with CAS number (CAS Registry Number 53123-88-9) was dissolved in DMSO as vehicle. Mice received a daily intra-peritoneal injection of Rapamycin (8 mg/kg) or vehicle after the CSDS for 7 consecutive days. 

### 2.6. Tissue Dissection and Dissociation

Mice were anaesthetised, and the brain was quickly dissected. The hippocampus was transferred into ice-cold oxygenated artificial cerebrospinal fluid (CSF: 87 mM NaCl, 2.5 mM KCl, 1.25 mM NaH_2_PO_4_, 26 mM NaHCO_3_, 75 mM sucrose, 20 mM glucose, 1 mM CaCl_2_, 2 mM MgSO_4_) and the left dorsal, right dorsal, left ventral and right ventral parts were dissected. The tissue was digested by the Papain Dissociation System Kit protocol (Worthington, Lakewood, USA) with some modifications. The papain buffer was supplemented with 10% trehalose (Sigma, Darmstadt, Germany), as described by Campbell and colleagues [28]. After trituration with a fire-polished Pasteur pipette and centrifugation at 300× *g* for 5 min, the gradient was made with Earle’s Balanced Salt Solution (EBSS) with 10% fetal bovine serum (FBS) used in place of ovomucoid protease inhibitor solution. The final cell pellet was resuspended in PBS buffer, filtered with a 30 µm pre-separation filter and kept at 4 °C for the cell sorting.

### 2.7. Cell Collection

The cells were sorted on a fluorescence-activated cell sorter BD FACSAria^TM^ III, using FACSDiva^TM^ Software (v8.0.2) (BD Biosciences, Heidelberg, Germany). Forward scattering and side scattering excluded debris, dead and doublet cells. Living cells were sorted into Neurobasal medium (21103049, Gibco, Darmstadt, Germany) supplemented with B27 (17504044, Gibco) and N2 supplements (17502048, Gibco). Cells were frozen at −80 °C for 4 h and then transferred in liquid nitrogen for 2 days in complete neurobasal medium supplemented with 10% DMSO.

### 2.8. Single-Cell Sequencing

For the scRNA-seq, three mice per group were analysed. After the brain dissection and cell dissociation, the living cells were isolated on a fluorescence-activated cell sorter, and the cells were frozen into the appropriate medium for two days. The cells were thawed and centrifuged at 4 °C, 300× *g* for 10 min. The cell pellets were resuspended into PBS, and the cell viability was measured using an automated cell counter. Cell numbers were diluted at the density of 1000 cells µL^−1^. For every sample, 3500 cells were loaded into a Chromium Single Cell 3′ Chip v2 (10X Genomics, Pleasanton, USA) and processed by following the manufacturer’s instructions. Paired-end sequencing (26/8/0/98 bp) of the libraries was performed onto the NovaSeq 6000 platform (Illumina, San Diego, USA) by using a S2 flowcell.

### 2.9. Bioinformatic Processing

The data were provided in demultiplexed form by the sequencing centre. Subsequently, Cellranger (v3.0.2) count was run on each sample with the additional option force-cells 2000. Next, Seurat (v3.1.0) was employed to filter and cluster the data using filter criteria of (<15% mitochondrial genes and between 200 and 2500 features; Appendix A). No technical issues or batch effects between the samples or groups were observed (Appendix A). Then, the data were log-normalised with a scaling factor of 10,000. Next, variable features were selected with a cut-off of 2000 features. After filtering criteria, the left and right parts for the dorsal and ventral hippocampus were computationally pooled together to form one sample for the downstream analysis (Appendix A). For each sample, 4000 single-cell were captured and analysed. On average, for each cell, 210,000 reads and 750 UMI were counted, 500 genes were detected and the sequencing saturation reached 95% (Appendix A). For initial clustering, 40 dimensions were chosen to be most optimal. This choice was based on the prediction using Elbow Plot, with a resolution parameter of 1 and the Jackstraw Plot. Both plots proved the significance of the chosen dimensions. Cluster stability was reassessed by choosing different resolution parameters and visualizing them with clustree (v0.4.3) (Appendix A). The number of cells per detected cluster is given in Appendix A. All 41 clusters were consistently identified in each group of mice with only slight differences in the number of cells for some specific sub-clusters (Appendix A). The obtained results were validated and cross-checked by additionally applying the recently developed non-parametric entropy-based Scalable Probabilistic Analysis framework (eSPA) [29,30], allowing a purely data-driven simultaneous solution of feature selection and clustering problems (Appendix A).

Each cluster’s top 10 marker genes were queried using the *FindAllMarkers()* function and subsequently used as input for panglaodb (https://panglaodb.se/, accessed on 12 October 2020) to facilitate defining the cell type. This approach was cross-validated using the R-Package SingleR (v1.2.4). 

Differential expression analysis was performed using the *FindAllMarkers()* with default settings. Genes with an absolute log fold change >0.25 and a false-discovery rate adjusted *p*-value < 0.05 were considered to be differentially expressed between experimental conditions. The over-representation of gene ontology (GO) terms in lists of differentially expressed genes from comparisons of experimental groups was tested with *the goana()* and *topGO()* function from the limma package (v3.4.4.3). 

### 2.10. Predicted Ligand-Receptor Interaction Analysis

A custom R script was used to assess putative interactions between different cell types based on a manually curated list of 2033 ligand-receptor interactions in the mouse downloaded from the CellTalkDB [31]. First, all upregulated genes, expressed in one of the sub-clusters of the four central microglia, oligodendrocytes, endothelial, and mural cell clusters, were extracted in Int animal’s dorsal hippocampus relative to control mice. Then, we filtered the list of DEGs for ligands and receptors occurring in the reference data set. Finally, we built all possible ligand-receptor pairs and kept only those pairs also present in the reference data set from CellTalkDB. A cell-cell interaction was defined as an upregulated ligand in cell type *x* having an upregulated interacting receptor partner in cell type *y*. The results were visualized as circos plots using the circlize R package v0.4.12, as well as directed graphs created with the igraph package v1.2.6. 

### 2.11. Statistical Analysis of Behavioural Data

Behavioural data were represented with boxplots including median, 1st quartile, 3rd quartile, interquartile range, and individual values. Unsupervised clustering with Gaussian mixture models (GMM) as implemented in the mclust R package v5.4.6 was employed to assign animals exposed to CSDS to three behavioural groups based on the SI test scores. Clusters were visualized in reduced multivariate space using principal component analysis on the mean centred and scaled behavioural data (all variables used for the PCA are show in Appendix A). For each principal component (PC), the loadings of the original variables corresponding to correlations between PCs and standardized input variables were calculated by calculating the matrix of eigenvectors with the diagonal matrix of square-root eigenvalues from the PCA. High loading values indicate behavioural measures most strongly contributing to the observed pattern in the PCA. In our analysis, animals corresponding to different stress groups were separated along the first PC, and the variables explaining this pattern most strongly were the social interaction index, but also the time spent in the interaction zone with the mouse present and the modified social interaction index with the CD1 mouse (Appendix A). Differences between behavioural sub-phenotype groups were statistically evaluated with a one-way ANOVA followed by Tukey’s post hoc test or Kruskal–Wallis test followed by Dunn’s post hoc test with a false discovery rate *p*-value adjustment. Assumptions of normality of residuals and homogeneity of variance were ascertained by inspecting Q-Q plots and residuals vs. fitted values plots, respectively. Modified social interaction scores were compared with linear mixed effects models by fitting random intercepts to each animal using the lme4 R package v1.1-23. Post-hoc comparisons of estimated marginal means were facilitated with emmeans v1.5.1. MSI scores were log_e_-transformed for statistical testing to satisfy the criteria of normality of residuals and variance homogeneity. The proportions of animals assigned to the three different behavioural phenotypes in the scRNA and pharmacological treatment experiments were compared with Chi-squared tests for independence. Differences between groups in the SI test after treatment with Rapamycin were statistically evaluated with a Two-way ANOVA followed by Tukey’s post hoc test. Unless otherwise stated, all *p*-values are two-tailed and a *p*-value < 0.05 was considered statistically significant.

## 3. Results

### 3.1. Characterisation of Distinct Behavioural Outcomes after CSDS

To understand cellular mechanisms underlying stress resilience, we performed a behavioural characterisation and classification of adult male C57BL/6J mice, after 10-day CSDS [27] followed by 4 days of rest and then a SI test (Figure 1a). To focus on strongly pronounced resilient and vulnerable phenotypes [10], we further investigated defeated animals with indices in the SI test of <0.75 and >1.15 (Figure 1b). We observed considerable heterogeneity in mice with SI index < 0.75 compared to the group with an SI index > 1.15 (variance ratio = 7.08, *p* = 0.003). To obtain more homogenous groups of animals that would be classically assigned to the susceptible phenotype, we decided to sub-cluster the defeated animals based on their SI performance by using an unsupervised Gaussian mixture model (GMM) clustering with three components. Moreover, we conducted further in-depth behavioural characterisation by applying the recently established MSI test [22]. Our clustering approach managed to produce three distinct groups of defeated mice which were separated along the first principal component (PC1) in reduced multivariate space (Figure 1c). Interestingly, even though we clustered mice only based on their performance in the SI test, we also observed differences between the three groups in the MSI test. The first group, corresponding to resilient (R) mice, showed behaviour similar to the control mice in the SI test (R mice median SI index = 1.3, control mice SI index = 1.2, *p* = 0.157; Figure 1d) and in the MSI test (R mice with CD1 median SI index = 1.15, control mice with CD1 SI index = 1.47, *p* = 0.933; R mice with 129/Sv median SI index = 0.77, control mice with 129/Sv SI index = 1.54, *p* = 0.251; Figure 1e). The second group, corresponding to a susceptible (S) phenotype, consisted of mice with the lowest SI index, i.e., below 0.5 (Figure 1d). Furthermore, in the MSI test, S mice showed significantly decreased interaction with both the CD1 strain (S mice with CD1 median SI index = 0.53, control mice with CD1 SI index = 1.47, *p* = 0.033; Figure 1e) and the unfamiliar 129/Sv strain (S mice with 129/Sv median SI index = 0.8, control mice with 129/Sv SI index = 1.54, *p* = 0.027; Figure 1e) compared to non-stressed animals. Finally, the third group produced by our GMM model consisted of animals that showed a susceptible phenotype in the SI test, but with average SI index that was still significantly higher than that of S animals (median SI index = 0.67, *p* = 0.025 vs. S mice; Figure 1d). Moreover, in the MSI test, these mice demonstrated features of resilient behaviour as indicated by a significantly increased selective interaction with the unfamiliar 129/Sv strain compared to the aggressor CD1 mice (CD1 median SI = 0.71, 129/Sv SI = 1.34, *p* = 0.008; Figure 1e), a feature previously reported [21]. Since this third group of mice exhibited a less pronounced susceptible phenotype as indicated by their performance in the SI test, we assigned the mice to an intermediate (Int) phenotype. Based on the notion that CSDS and behavioural consequences may alter cognition capacities, a Y-maze test was performed. Even though we observed a decrease in distance moved during the habituation phase for the resilient mice, the lack of differences in the novel arm preference suggests the social avoidance we observed is not associated with anxiety-like phenotypes or cognition capacities (Appendix A). Similar observations have already been reported between CSDS and other anxiety tests, i.e., elevated-plus maze and open-field tests [32,33,34]. These studies and our observation suggest that social avoidance and anxiety-like phenotypes are controlled by different neuronal circuits/molecular mechanisms. To investigate whether the distinct behavioural outcomes we observed were related to region and cell-type specific transcriptional changes in the hippocampus, we selected three mice from each group with the most pronounced and consistent phenotype in the SI and MSI tests for further transcriptomic analysis (Figure 1c–e, filled dots, and triangles).

### 3.2. Single-Cell RNA-Seq Analysis Identifies 41 Distinct Cell Clusters in the Hippocampus

To uncover the hippocampal cellular populations actively involved in stress resilience, we performed single-cell RNA-seq followed by an extensive computational analysis of the selected mice (Figure 1a). Out of the 32,000 single cells captured, 29,358 cells were kept after the filtering, without enrichment in any hippocampal sub-region or group of mice (see Methods and Appendix A). To identify different brain cell populations, we employed unsupervised graph-based clustering and revealed 41 distinct clusters, visualised with the uniform manifold approximation and projection (UMAP) dimensionality reduction technique (see Methods, Figure 2a and Appendix A). Each cluster was annotated using a combination of the top 10 genes per cluster and well-known markers for astrocytes, neural stem cells, neurons, mural cells, endothelial cells, oligodendrocyte precursor cells (OPC), oligodendrocytes, and microglia. A dot plot (Figure 2b, Appendix A) and UMAP dimensional reduction (Figure 2c) were employed to visualise the cell-type specific marker genes. As reported by others, the percentage of each cell type does not necessarily reflect the actual proportion of the different cell populations in the adult mouse hippocampus, mainly due to their different sensitivity to the dissociation protocol employed [35,36]. Due to the low number of neurons detected, stress resilience neuronal molecular mechanisms are largely missing from our analysis. However, the focus on non-neuronal cells, representing 50% of the brain cell population, allowed us to identify unexpected and promising mechanisms underlying stress resilience. 

### 3.3. Specific Cell-Type CSDS Responses in the Defeated Mice

To identify the hippocampal sub-region and the cell types actively involved in stress resilience, we determined the differentially expressed genes (DEGs) for each group of defeated animals compared to control mice. Surprisingly, by combining all the cell types to mimic a bulk RNA-seq, we detected the highest number of DEGs in the dorsal hippocampal sub-region for the Int animals and to a smaller extent for the S mice (Figure 2d). In line with this, more differences were observed in the individual brain cell types. Indeed, the Int mice showed the highest transcriptomic difference in the dorsal hippocampal sub-regions and also in every cell type captured, i.e., microglia, oligodendrocytes, endothelial and mural cells, suggesting a molecular CSDS hippocampal response (Figure 2e–h). The R mice showed a similar transcriptome as control mice in all the cell types isolated (Figure 2e–h). Surprisingly, in the S mice, very few DEGs were observed (Figure 2e–h). In the ventral hippocampus, few or no DEGs were detected for any comparison and cell type, suggesting that non-neuronal cells, in this particular region, were not involved in CSDS resilience (Figure 2e–h). These observations suggest that mainly the non-neuronal cells in the dorsal hippocampus are actively involved in the CSDS response. Moreover, a parameter-free clustering approach also observed this regional-dependent cell cluster signature; supporting our observations (Appendix A). The lack of transcriptomic differences in the R and S mice compared with previous bulk analyses could be explained by (i) the lack of the neuronal cell population in our experiment, (ii) the time point where the samples were collected as already reported by Bouvier et al. [37], and (iii) the sub-classification of the classical heterogeneous susceptible mice, suggesting that the susceptible signature is mainly present in the intermediate sub-group only.

An analysis of the differentially expressed genes between the three groups of defeated animals for each cell type revealed that the highest number of genes was also observed in the dorsal hippocampus of Int mice compared to R mice (Appendix A). Furthermore, the number of genes, about 350 genes, was similar between cell types and consistent with the comparison to the control group. Interestingly, around 100 DEGs were observed in the ventral hippocampus of R mice compared to S mice for microglia, oligodendrocytes and mural cells. However, further analysis revealed that these DEGs were observed in only one sub-cluster for each of these cell types, which might not reflect the molecular mechanism in the entire cell population (Appendix A). Finally, the DEGs in the dorsal hippocampus of Int mice compared to the S mice were due to the differences in the Int group. Therefore, we decided to focus the rest of the analysis on the comparison of the three groups of defeated mice with the control group.

To corroborate the single-cell analysis with the behaviour phenotype, we analysed the differential expression of immediate early genes known to be activated by stress. We observed that Arc, Egr1, c-Fos, and Jun genes were significantly overexpressed in the dorsal hippocampus in the Int mice compared to the control mice (Figure 2i). Surprisingly, these genes were slightly downregulated in the R mice (Figure 2i). However, an overexpression of CSDS markers [38] was observed in all the three groups of defeated animals with a strongest and highest overexpression in the hippocampus of S mice, where the highest number of cell types expressing these markers in every hippocampal sub-region was observed (Figure 2j). These CSDS markers were also overexpressed in the dorsal hippocampus of Int mice, whereas they were overexpressed in only few cell clusters in the ventral hippocampus of R mice compared to control (Figure 2j). Taken together, these observations showed that the three groups of defeated mice feature distinct transcriptomic signatures at the cell type level as compared to the control mice. Moreover, the involvement of the dorsal hippocampus in CSDS response is manifested for every non-neuronal brain cell type analysed. The Int mice showed the strongest transcriptomic difference compared to control mice, suggesting an ongoing molecular CSDS-induced hippocampal response occurring in this specific group. 

### 3.4. The Microglia in the CSDS Response 

Microglia, the resident brain macrophages, migrate to the brain during embryonic brain development, where they self-renew, and are involved in several processes such as neuronal development and synapse pruning. In adulthood, microglia maintain brain homeostasis and neuronal environment [39,40]. To highlight microglia’s role during CSDS response, we identified the DEGs and their related biological pathways for the microglia sub-clusters between the defeated groups and the control mice. We observed the highest number of DEGs in the dorsal microglia sub-cluster 1 of Int mice (269 genes) compared to the control mice (Figure 3a).

In contrast, in the R and S mice, the highest quantity of DEGs was observed in the ventral microglia sub-cluster 1 with 50 genes, and in the ventral microglia sub-cluster 2 with 20 genes, respectively (Appendix A). The DEGs in the dorsal hippocampus microglia of the Int mice were involved in several biological pathways, including immune and stress response. In contrast, we did not detect overrepresented biological processes in R and S mice due to the low number of DEGs observed (Figure 3b). Similar to the peripheral macrophage polarization, two pathways of microglia polarization have been described: (i) the neurotoxic M1 and (ii) the alternative neuroprotective M2 pathways known to express distinct markers, e.g., (i) Il-1β, Il-6, TNF-α, and (ii) Il-10, TGF-β, Socs3, respectively [39,40]. A strong overexpression of M1 and M2 markers was detected in the dorsal part of Int mice (Figure 3c). Surprisingly, the R mice showed a slight downregulation of a few M1 and M2 markers in the ventral hippocampal sub-region. In contrast, mild overexpression of only M1 markers in the dorsal hippocampus was observed in S mice (Appendix A). Strikingly, altogether, although markers were overexpressed, we detected fewer microglial cells in the Int mice expressing these markers compared to the control group in the dorsal hippocampal sub-region (Figure 3d). Similarly, fewer cells expressed the M1 markers in S mice versus control, whereas a slightly higher percentage of cells expressed the M1 and M2 markers in the R mice relative to controls (Appendix A). These observations suggest that the microglial cells contain a basic level of M1 and M2 gene expression, enabling a fast cell activation during or after stress or inflammation. Furthermore, our observations indicate that only a small number of activated microglial cells is enough to induce an active molecular stress response in the hippocampus. 

### 3.5. The Oligodendrocytes in the CSDS Response

The oligodendrocytes are the myelinating cells in the central nervous system, derived from the OPC, and involved in axon ensheathment, which regulates neuronal activity. For each stage of oligodendrocyte differentiation and maturation, specific markers are well described. In adulthood, a decrease of oligodendrocytes and an impairment of remyelination have been associated with disorders such as multiple sclerosis and schizophrenia [41,42]. To highlight the role of the oligodendrocytes in stress resilience, we identified the DEGs in each oligodendrocyte sub-cluster for each group of defeated mice compared to control mice. In the oligodendrocyte sub-clusters 1 and 2 of the dorsal hippocampus of Int mice, 251 and 253 DEGs, respectively, were observed (Figure 4a). Similar to the microglia results, fewer DEGs were present in the oligodendrocyte sub-clusters of R and S mice compared to control mice, with the most-significant differences solely in the ventral hippocampus (Appendix A). The DEGs from the dorsal hippocampus of Int mice were involved in several pathways, including axon ensheathment, and (re) myelination (Figure 4b). An overexpression of OPC and oligodendrocyte differentiation and maturation markers were detected in both hippocampal sub-regions of Int mice with a more substantial overexpression in the dorsal part (Figure 4c). In contrast, these markers were not differentially expressed in R and S mice relative to controls (Appendix A). These observations suggest that CSDS induced an active transcriptional response related to remyelination in the Int mice only. Surprisingly, among the dorsal oligodendrocyte DEGs in the Int mice, some genes are reported to be involved in the biological pathways of microglia cell activation involved in the immune response. The genes underlying this pathway are ligands and receptors known to induce an immune response, including C1qa, Ccl3 and Ccl5 [43]. These genes were upregulated in the dorsal hippocampal sub-region of Int mice but not in the R and S mice (Figure 4d). The C1q complex is known to induce macrophage M2 polarization [44], which is in accordance with the M2 microglia activation observed above (Figure 3c). Moreover, the DEGs from the microglia sub-cluster 8 in the dorsal hippocampus of Int mice were also involved in positive glial cell differentiation pathways (Figure 4e). The genes underlying this pathway are microglia-secreted factors and receptors, including Lgals3, Lpl, and Cd74, known to be involved in OPC proliferation and differentiation, as well as remyelination processes [45,46]. These markers were overexpressed in the dorsal hippocampal sub-region of Int mice (Figure 4f). In contrast, no differential expression was detected in the R mice (Appendix A). Taken together, we detected a positive feedback loop between the oligodendrocytes and microglia in the dorsal hippocampus of Int mice on the transcriptional level that could lead to neuroprotection and/or neurogenesis in the Int mice upon CSDS (Figure 4g). In the S mice, remyelination factors were overexpressed by microglial cells. However, in this group of mice, only one gene involved in the remyelination was detected compared with the Int mice (Appendix A). Therefore, the oligodendrocytes from S mice were presumably not able to secrete, in return, the M2 microglia activation-ligands (Appendix A). 

### 3.6. The BBB in the CSDS Response

The neurovascular unit comprises endothelial cells surrounded by pericytes, vascular smooth muscle cells, perivascular astrocytes, microglia and oligodendrocytes [47,48]. The pericytes and vascular smooth muscle cells form the mural cell population and contribute to BBB maintenance [49]. Analysis of the DEGs revealed that the neurovascular unit occupies a crucial role in the CSDS response within the dorsal hippocampus (Figure 2g,h). In the endothelial and mural sub-clusters 1 and 4 of the dorsal hippocampal sub-regions of Int mice, the highest number of DEGs was observed with 351, 258, 312, and 107 genes, respectively (Figure 5a,e). Similar to our previous observations, only few DEGs in the endothelial and mural cells were detected in the R and S mice compared to control mice (Appendix A). The DEGs from the neurovascular unit cells in the dorsal hippocampus of Int mice were involved in several biological pathways, including system development and developmental processes (Figure 5b,f) and more specifically in brain development, angiogenesis, neurogenesis and nervous system development (Figure 5c,g). After CSDS, leakage of the BBB in the Nucleus accumbens and in the hippocampus due to a decrease of claudin-5 expression had been identified previously [17]. In line with this, we detected an overexpression of several angiogenic growth factors (Egfl7, Pdgfα, Pdgfβ), receptors (S1pr1, Robo4, Anxa2), a matrix metalloproteinase inhibitor (Ecm1), as well as cell-cell adhesion and gap-junction molecules (Jam3, Gja4) in the neurovascular unit cells in the dorsal hippocampus of Int mice, promoting angiogenesis, restoration, and maintenance of the BBB (Figure 5d,h). Only a few of these genes were also overexpressed in S mice, whereas no differences were seen in the R mice (Appendix A). Furthermore, the endothelial and mural cells in the dorsal hippocampus of Int mice overexpressed secreted factors (Bmp4, Edn3, Btg2, Serpine2), extra-cellular matrix components (Col4α1, Col4α2, Lamβ2), cell-cell interaction proteins and gap junction proteins involved in neurogenesis and neuron projection development pathways (Figure 5d,h). Indeed, Bmp4 by interaction with Tgfβr, Btg2, and Serpine proteins were reported to induce neuronal differentiation and neurite extension [50,51,52]. Secretion of extra-cellular matrix components by pericytes promotes OPC proliferation and differentiation, which is necessary to induce and maintain axon ensheathment [53]. None of these genes were differentially expressed in the R and S mice compared to control mice. Taken together, endothelial and mural cells, which compose the neurovascular unit, were involved in CSDS response in Int mice, suggesting an adaptation process leading to restoration of BBB integrity and promotion of neurogenesis and axon ensheathment.

### 3.7. Cell-Cell Interaction after CSDS 

Our analysis revealed a complex stress response in the dorsal hippocampus of Int mice. Therefore, we analysed the putative ligand-receptor interactions between the cell types based on the upregulated genes in Int mice. A circos plot revealed that multiple ligands and their respective receptors were overexpressed by every non-neuronal brain cell type isolated, leading to a dynamic and complex putative cell-cell interaction network in the dorsal hippocampus of Int mice in response to CSDS (Figure 6a). We observed both cell-type-specific connections as well as interactions between the different brain cell types. Altogether, we detected 68 unique ligand-receptor pairs upregulated in the dorsal hippocampus of Int mice (Figure 6b). Some of the interactions were cell-type specific, such as the Fgf1-Fgfr2 pair, inducing oligodendrocyte proliferation. However, the majority of the ligand-receptor interactions involved the different brain cell types (Figure 6b). Indeed, the ligands of the Ccr5 microglia receptor, i.e., Ccl3, Ccl4, Ccl5 and Ccl7, were overexpressed by microglia, oligodendrocytes, endothelial cells and mural cells, and these interactions induce M2 microglia polarization (Figure 4d and Figure 6b). Furthermore, the microglia-overexpressed serpine1 putatively binds to the oligodendrocytes-overexpressed receptor Lpr1b to promote OPC proliferation and differentiation (Figure 6b). Importantly, the upregulated genes in the dorsal hippocampus of R and S mice relative to controls did not overlap with any ligand-receptor pairs included in the CellTalkDB reference set (http://tcm.zju.edu.cn/celltalkdb/, accessed on 14 January 2021), indicating that the sophisticated interaction network between the different cell types was not present in these mice. Taken together, we uncovered complex, dynamic and under-investigated transcriptional processes involving non-neuronal cells in the dorsal hippocampus of Int mice, suggesting an adaptive molecular CSDS response occurring exclusively in this group of defeated mice.

### 3.8. Rapamycin Drug Treatment Improves Stress Resilience

Our data suggest a critical role of the glial-neurovascular unit in the active molecular response after CSDS. The detection of a significant increase of expression of genes involved in immune and stress response as well as of M1 and M2 microglia markers in the dorsal hippocampus of Int animals, together with an overexpression of Ccr5 receptor ligands in microglia, points towards a crucial importance of microglia and microglia polarisation in the adaptation process (Figure 6b). Moreover, the microglia M2 polarisation seems to be actively involved in a positive feedback loop with the oligodendrocytes promoting oligodendrogenesis and therefore myelination in Int mice only. Finally, in endothelial and mural cells of the dorsal hippocampus of Int but not of R or S animals, overexpression of matrix metalloproteinase 3 and 9 (MMP3, MMP9) inhibitors [54,55], i.e., Ecm1 and Timp3 (Figure 6c), was observed. This overexpression in Int mice suggests that the integrity of the BBB plays a key role [56]. In line with this, overexpression of ECM components, e.g., Col4α1, Col4α2 and Lamβ2, which influence the BBB [57], was also detected in endothelial and mural cells of the dorsal hippocampus of Int mice compared to control mice (Figure 6c). Again, none of these genes were differentially expressed in the R and S mice, respectively, compared to control mice. Altogether, these observations guided us to a common regulator of these above-mentioned signaling pathways, the mTOR kinase, which is a crucial effector of cerebrovascular functions, regulating neurogenesis, myelination, synaptic plasticity, microglia activation and polarization and influencing the permeability of the BBB and the composition of the ECM [58,59,60,61,62,63,64,65]. The mTOR pathway activation relies on the phosphorylation of several proteins, a post-transcriptional modification that cannot be detected in transcriptomic analysis. However, the downstream effects of the mTOR pathway share remarkable similarities with all the molecular processes we observed in the dorsal hippocampus of Int mice. Moreover, KEGG pathway analyses revealed that the negative regulators of the mTOR complex, i.e., the MAPK and PI3K signaling pathways, were upregulated in the microglia and endothelial cells of the dorsal hippocampus of Int mice as compared to control (Appendix A). In line with this, mTOR inhibitors, such as rapamycin, were previously shown to upregulate the PI3K-Akt and MAPK pathways [66]. Therefore, we compiled strong evidence suggesting that inhibition of the mTOR kinase with rapamycin would activate the above identified mechanisms in the entire group of susceptible mice, i.e., with a SI index < 0.75, and thereby improve stress resilience behavior. To this end, mice were treated with the mTOR inhibitor rapamycin for 7 days immediately following the CSDS, and then the SI test was performed. In the vehicle group, the defeated mice showed significantly lower SI indices compared to the control mice (defeated mice SI index = 0.64, control mice SI index = 1.04, *p* = 0.0021; Figure 6d). On the contrary, in the rapamycin treated group, similar SI indices were observed in the defeated and in the control mice (defeated mice SI index = 1.07, control mice SI index = 1.08, *p* = 0.9993, Figure 6d). In the defeated animals, a significant increase of the median SI index in the rapamycin group compared to the vehicle group was observed (defeated mice rapamycin group SI index = 1.07, defeated mice vehicle group SI index = 0.64, *p* = 0.0010, Figure 6d). Finally, to analyze the effect of the treatment on the stress resilience group size previously identified, the animals were reassigned to the three stress resilience groups, i.e., R with SI index > 1.15, S with SI index < 0.50 and Int with SI index between 0.50 and 0.75, similarly to Figure 1d. In the two independent experiments we performed, i.e., for the single-cell RNA-seq and for the pharmacological intervention, we compared the proportion of stressed animals falling in one of the three categories (Figure 6e). No differences in the subgroup proportions were observed between the scRNA experiment and the vehicle group of the pharmacological intervention experiment, suggesting that our behavioural stratification is robust over independent experiments. In the rapamycin treated group, a shift from the susceptible group to the resilient group was observed compared to the vehicle group (Figure 6e). Indeed, in the rapamycin group, almost 75% of the animals are resilient and only one mouse was classified as susceptible with a SI index of 0.497, whereas in the vehicle group, 83% of the animals were susceptible. By comparing the scRNA experiment with the rapamycin treated group, a similar trend toward the resilience group was also observed (*p* = 0.08) (Figure 6e). Interestingly, the group size shift was observed only in the R and S groups, whereas the size of the Int group remained stable over the experiments, suggesting that the intermediate group represents indeed an intermediate step from the susceptible behavior to the resilient behavior. Altogether, these data revealed that inhibition of mTOR with rapamycin can indeed induce stress resilience.

## 4. Discussion

Stress resilience is a multifactorial process involving both brain and peripheral mechanisms, including the immune system and gut microbiome [4,8,9]. An additional level of complexity is given by the diversity of cell types affected and their intricate interactions. To uncover the impact of cellular heterogeneity upon CSDS and to reveal molecular signatures of the different hippocampal cell types, we took advantage of a single-cell transcriptome analysis. Our fine-grained analysis revealed major transcriptional changes in the glial and neurovascular systems after CSDS. Among the processes identified, we highlighted that specific microglial subpopulations are actively involved in a positive feedback loop with oligodendrocytes, which could potentially promote remyelination after CSDS. Moreover, we highlighted the central importance of the neurovascular system in the beneficial molecular response to CSDS. In support of our observations, we found that inhibition of the mTOR kinase, known to regulate several of the mechanisms we revealed, after the stress substantially improved resilience outcomes.

CSDS has been widely used as a behavioural paradigm to elucidate stress resilience mechanisms. The original paradigm followed by the SI test identified two groups of defeated mice, resilient and susceptible, respectively, exhibiting specific transcriptomic responses in various brain regions [10,11,12,13]. Recently, replacing the commonly used SI test by a MSI test identified, within the susceptible group, a third group of mice exhibiting phenotypic characteristics of resilience, in particular the ability to discriminate aversive from non-aversive stimuli [21,22]. In our present study, we combined both approaches to outcome measures by performing both SI and MSI tests after CSDS, to obtain a more comprehensive behavioural dataset. We used an unsupervised clustering approach to produce three defeated sub-groups that shared behavioural similarities with the three sub-groups of defeated mice previously reported [21], which we named here resilient (R), intermediate (Int), and susceptible (S). Subdividing defeated animals into three groups instead of the well-established dichotomous approach in combination with a fine-grained scRNA-seq dataset, revealed novel insights into the complexity of stress resilience mechanisms. Indeed, although the CSDS markers were overexpressed in the ventral hippocampus in the resilient group of defeated mice (R, SI > 1.15), social avoidance was absent. Unlike previous analyses performed 2 days after the stress [10,11,12,13], we did not detect molecular changes 10 days after stress indicating an active stress resilience process, but rather suggesting these animals either displayed a passive resilience type [4] or have already recovered from the stress [37]. In the remaining two groups of defeated mice, which would be classically defined as susceptible mice (SI < 1.0), behavioural and molecular distinctions were uncovered. In the Int mice, we observed a less pronounced defeated phenotype, with dynamic and active molecular changes in the dorsal hippocampal sub-region, indicating a more complex stress response at the behavioural and molecular level. In contrast, in the S mice, only weak molecular changes were observed, suggesting that these animals were susceptible and without active molecular mechanisms promoting stress resilience.

Previous transcriptomic analyses on, e.g., NAc and ventral hippocampus have reported clear transcriptomic changes in the CSDS resilient mice, implying that stress resilience is a dynamic and active molecular process [10,11,12,13]. However, the mice in these studies were classified as resilient or susceptible on the day after CSDS. We believe that stress resilience is a dynamic process requiring time to be established, and that the acute stress effect may have interfered with the SI test. We can extrapolate that some animals classified as resilient and susceptible, respectively, on the day after CSDS may have been classified into another group if the SI test had been performed at a later time point, as suggested in a previous longitudinal study [23]. Therefore, we performed the SI and MSI tests 5 and 8 days after CSDS, respectively, allowing the animals to establish their long-term stress response and get the adaptation program started. This difference in the behavioural paradigm and the lack of the neuronal cells in our analysis due to technical limitations would explain the absence of DEGs in our resilient animals compared to the previous bulk transcriptomic analyses. We cannot exclude that the existence of prior differences or neuronal mechanisms could have predisposed or led the animals to gain resilience without engaging non-neuronal molecular stress response. Moreover, we addressed the point of behavioural heterogeneity among the susceptible mice and identified a sub-group of classical susceptible mice that showed major transcriptional changes. In the previous transcriptomic analyses [10,11,12,13,18,19], no information regarding the SI index of the mice selected for the analyses was provided. Therefore, we cannot exclude that the signatures observed in the classical heterogeneous susceptible mice from previous bulk transcriptomic analyses originated from the intermediate sub-group, which drew our attention.

At the global transcriptomic level, we observed that the resilience-related changes in the gene expression pattern of the most relevant cell types were significantly more pronounced in the dorsal than in the ventral hippocampus. Although we also applied a second, completely different, parameter-free clustering approach, the reported region-dependent cell cluster signature remained impressively robust (Appendix A). 

The intermediate phenotype that we identified in our study coincided with transcriptional glia cell activation, i.e., microglia, oligodendrocytes in the dorsal hippocampus. Some studies have previously revealed that the intensity of the stress, the numbers of microglial cells, and the over-activation/over-inhibition of microglia throughout the brain lead to different psychiatric disorder outcomes [67]. However, these studies were focused on neurotoxic M1 microglia activation. Here, we detected the neurotoxic M1 and the neuroprotective M2 microglia activation in the Int mice following CSDS, allowing phagocytosis of dead cells and removal of myelin debris on the one hand and remyelination on the other, both of which are important players in regenerative processes [68]. We believe that the M1-M2 microglia balance is necessary to restore brain homeostasis and to establish long-term stress resilience. 

After CSDS, a reduction of myelination in the ventral hippocampus and mPFC has already been reported in susceptible mice [18,19,69]. Complementary to this, here, we highlighted an overexpression of genes related to oligodendrogenesis and myelin differentiation in the dorsal hippocampus of Int mice, which might indicate an active ongoing process potentially promoting CSDS resilience. Menard and colleagues reported a BBB leakage in the NAc and in the hippocampus, leading to peripheral interleukin IL-6 infiltration in the susceptible animals [17]. In line with this, we found factors and tight-junction proteins involved in BBB formation and maintenance to be overexpressed, suggesting that an active process of BBB restoration might be engaged in the Int mice.

Most importantly, the scRNA-seq analysis also implies a robust cell-cell communication through ligand-receptor interactions, secretions of factors, and extracellular matrix components between the non-neuronal cells, i.e., microglia, oligodendrocytes, endothelial, and mural cells in the dorsal hippocampus of Int mice. These interactions have the potential to activate several mechanisms, including M2 microglia polarisation, proliferation and differentiation of OPC into oligodendrocytes promoting myelination, neurogenesis, and BBB restoration (Graphical Abstract, Figure 6f). Glial cell interactions have already been reported in several studies. For example, in multiple sclerosis, CRYAB positive oligodendrocytes can come in contact with and activate microglia, whereas myelin debris removal by activated microglia is required for inducing remyelination [70,71]. Moreover, the anti-depressant-like properties of single vaccination with myelin basic protein peptides can be attenuated by an additional injection of pro-inflammatory cytokines, again suggesting that these different processes are interrelated [72]. Taken together, these findings strongly support a model where a coordinated involvement of glial cells is indicative of major stress-response adaptive processes. This involvement of glial networks could also further explain anatomical and neuronal changes previously documented. Indeed, the lack of activation of the mechanisms highlighted here could explain the decrease in volume, the reduction in apical dendritic length of CA1 and CA3 and the reduced number of GABAergic neurons observed in dorsal hippocampus of defeated, mainly susceptible, rats after chronic stress [73,74]. 

Notably, we found that in the intermediate group, the dorsal hippocampus, a brain region important for associative learning processes, is strongly involved. In fact, the reported resilience conducing effect of rapamycin may also include a learning mechanism, as rapamycin was shown to influence consolidation and re-consolidation of fear memory in the dorsal hippocampus [75,76,77]. This is also in agreement with the recent observation that CSDS-induced avoidance involves conditioned learning [22]. 

Our observation that rapamycin treatment given directly after CSDS increases both the individual SI index of defeated animals and the fraction of resilient animals within the defeated group suggests that rapamycin is a stress resilience drug candidate. This has promising implications for translation since it means that substances that inhibit the mTOR kinase could be used to boost resilience after trauma. However, additional studies are needed to elucidate the exact resilience-promoting effect of mTOR inhibitors.

## 5. Conclusions

To summarize, our fine-grained behavioural and transcriptomic analyses highlighted a sub-group of mice in the commonly classified susceptible group, showing active and dynamic non-neuronal molecular response associated with brain restoration and homeostasis and therefore presumably promoting the stress resilience outcome. Our results suggest that CSDS affects neural signalling processes, and homeostasis of the dorsal hippocampus, and that neuroprotective mechanisms from the non-neuronal cells could provide an active stress coping strategy. As a proof-of-concept approach, by treating animals with rapamycin, we further show that resilience can be strengthened after stress using pharmacological agents. We believe that the plethora of candidate mechanisms, as identified in our study, can pave the way for the development of targeted pharmacological interventions during or even better immediately after a period of stress to prevent psychological dysregulation and to promote stress resilience.

## Figures and Tables

**Figure 1 cells-11-03405-f001:**
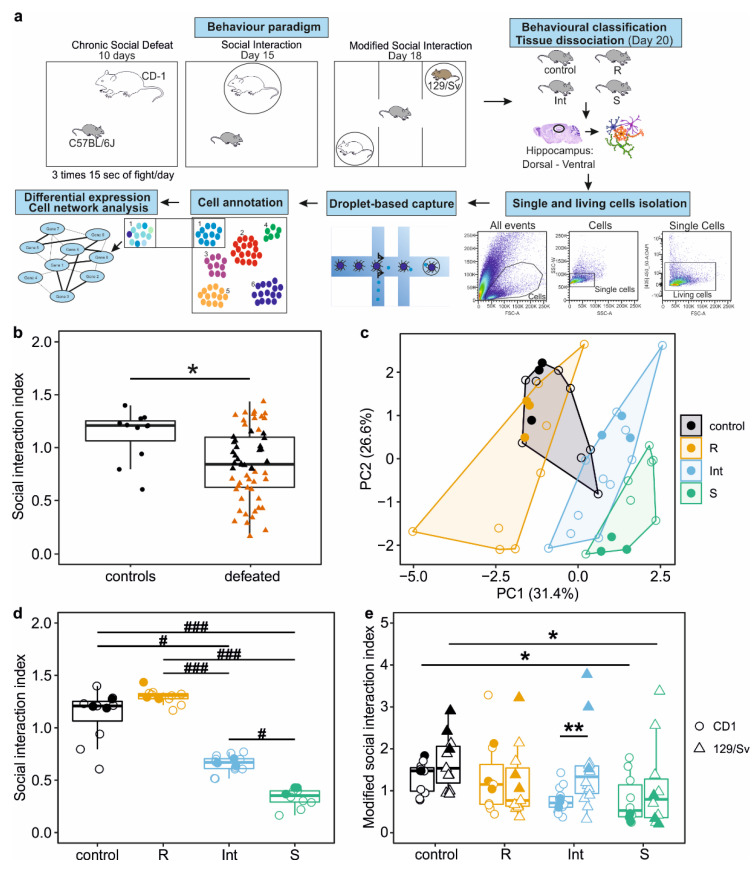
Behavioural classification of mice after chronic social defeat stress. (**a**) Overview of the experiments. Chronic social defeat stress was applied for 10 days with 3 times 15 s of fight per day. After 4 days, behavioural profiling was started. After the behavioural classification, three mice per stressed group (R, Int, S) and three non-stressed mice were selected. Brains were collected 2 days after the last behavioural test, and the dorsal and ventral hippocampus were dissected. Tissues were dissociated, and single and living cells were isolated by using a fluorescence-activated cell sorter. A total of 32,000 single-cells were captured by a droplet-based method and sequenced. Different cell populations were identified, the differentially expressed genes between the groups were analysed, and cell networks were uncovered. (**b**) Social interaction (SI) index of controls and defeated mice. After CSDS, two subgroups of defeated mice (highlighted in brown), i.e., the socially avoiding (SI < 0.75) and the socially non-avoiding (SI > 1.15) mice, were further subjected to a modified social interaction (MSI) test. (**c**) Based on their SI scores, the selected defeated mice highlighted in Figure 1b were clustered into three sub-phenotypes. Behavioural groups are visualized with principal component analysis, with four groups termed control (black), resilient (orange, R), intermediate (blue, Int), and susceptible (green, S) mice. The area around each cluster corresponds to the convex hull. (**d**) SI index of the four groups of mice. (**e**) SI index in the MSI test of the four groups of mice. Filled circles and triangles in (**c**–**e**): individual mouse taken for subsequent single-cell RNA-seq experiment. * *p* < 0.05, ** *p* < 0.01 (Wilcoxon test in b, linear mixed-effects model followed by Tukey comparisons of marginal means in (**e**), # *p* < 0.05, ### *p* < 0.001 (Kruskal–Wallis test followed by Dunn’s posthoc comparisons in (**d**)).

**Figure 2 cells-11-03405-f002:**
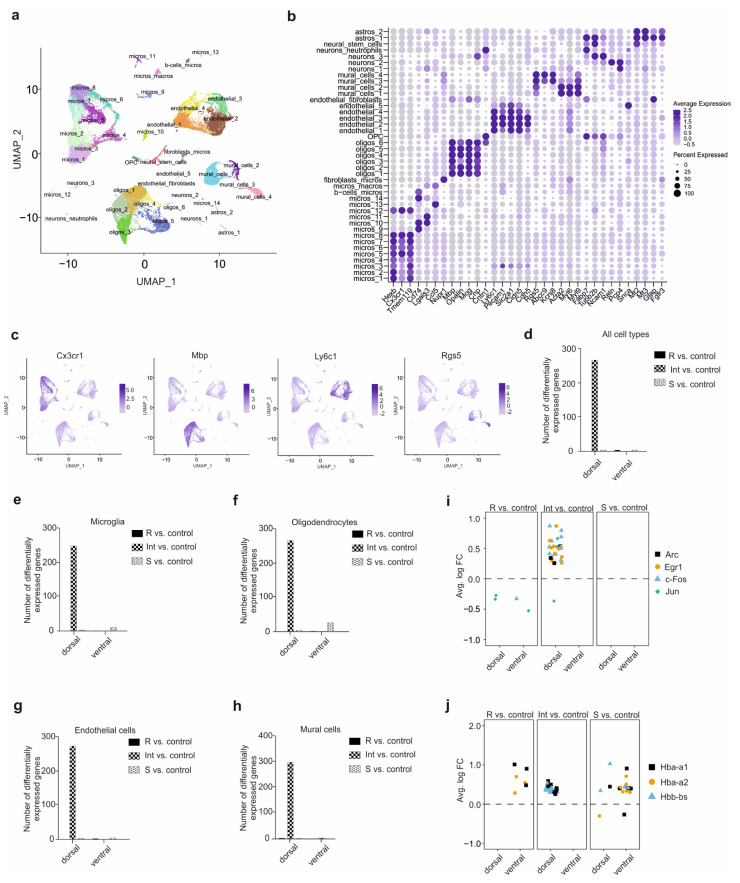
Single-cell analysis identifies different cell types in the different parts of the hippocampus as a specific response to chronic social defeat stress. (**a**) The uniform manifold approximation and projection (UMAP) plot of 29,358 cells in 41 clusters identified after thorough quality control. astros, astrocytes; OPC, oligodendrocytes progenitor cells; oligos, oligodendrocytes; micros, microglia; fibroblasts/micros, fibroblasts and microglia; micros/macros, microglia/macrophages; b-cells/micros, B lymphocytes/microglia. (**b**) Cell type annotation is based on the expression of well-known marker genes. The dot plot shows the expression of these genes across all the cell types. The dot size represents the percentage of cells expressing the gene, whereas the colour intensity of the dot corresponds to the average expression level. (**c**) The UMAP visualization of the 4 major cell populations, i.e., microglia, oligodendrocytes, endothelial and mural cells, showing the expression of representative well-known cell-type-specific marker genes. (**d**–**h**) Number of differentially expressed genes of the resilient (R), intermediate (Int), and susceptible (S) mice compared to control mice and among all cell populations (**d**), microglia (**e**), oligodendrocytes (**f**), endothelial cells (**g**), mural cells (**h**) within the full hippocampus (hippocampus) and within the different parts: dorsal and ventral. (**i**,**j**) Differential expression of the immediate early genes (Arc, Egr1, c-Fos, Jun) (**i**) and the chronic social defeat stress marker genes (Hba-a1, Hba-a2, Hbb-bs) (**j**) of the resilient (R), the intermediate (Int) and the susceptible (S) mice compared to control mice in the entire hippocampus (hippocampus) and in the dissected parts of the hippocampus (dorsal, ventral). Each dot represents the average log2 fold change of differential expression of the corresponding gene between the two groups of mice in a given cell cluster.

**Figure 3 cells-11-03405-f003:**
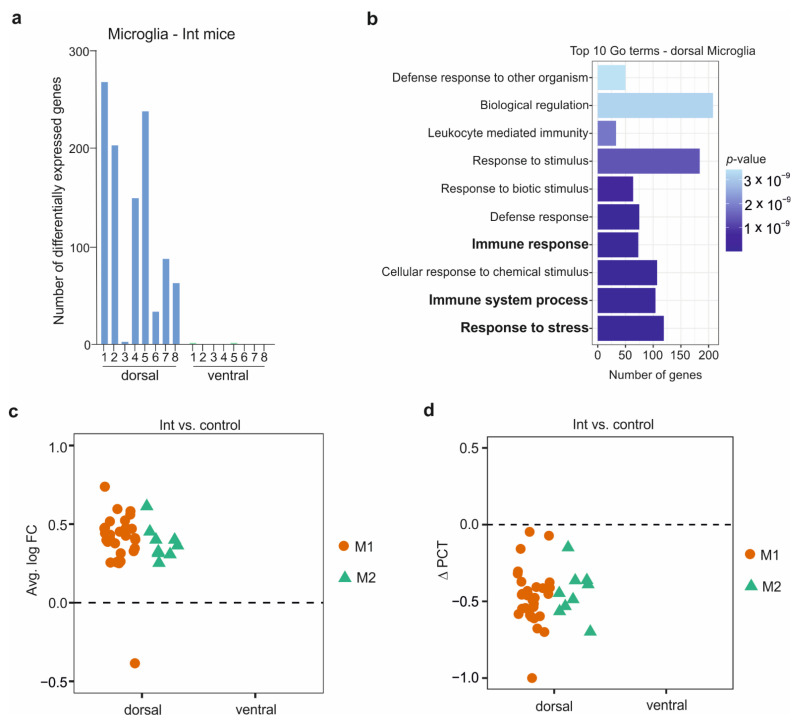
Chronic social defeat stress induces a microglia immune response in the hippocampal dorsal part of intermediate mice. (**a**) Number of DEGs among the microglial sub-clusters 1 to 8 in the intermediate mice compared to control mice in the dorsal and ventral parts of the hippocampus. (**b**) Top 10 GO terms analysis related to the DEGs identified in the dorsal hippocampus of the intermediate mice. The most relevant pathways are highlighted in bold. (**c**) Differential expression of M1 (Cxcl10, Fcgr2b, Fcgr3, H2-Aa, H2-D1, H2-Dmb1, H2-K1, H2-Oa, H2-Q4, H2-Q6, H2-T23, Tnfαip2, Tnfαip8l2, Il-1β) and M2 (Il-10rα, Socs3, Tgfβ1) markers in the dorsal and ventral parts of the hippocampus in the intermediate (Int) mice compared to control. (**d**) Difference in the percentage of cells (ΔPCT) expressing the M1 (Cxcl10, Fcgr2b, Fcgr3, H2-Aa, H2-D1, H2-Dmb1, H2-K1, H2-Oa, H2-Q4, H2-Q6, H2-T23, Tnfαip2, Tnfαip8l2, Il-1β) and M2 (Il-10rα, Socs3, Tgfβ1) markers in the dorsal and ventral parts of the hippocampus in the intermediate mice compared to control.

**Figure 4 cells-11-03405-f004:**
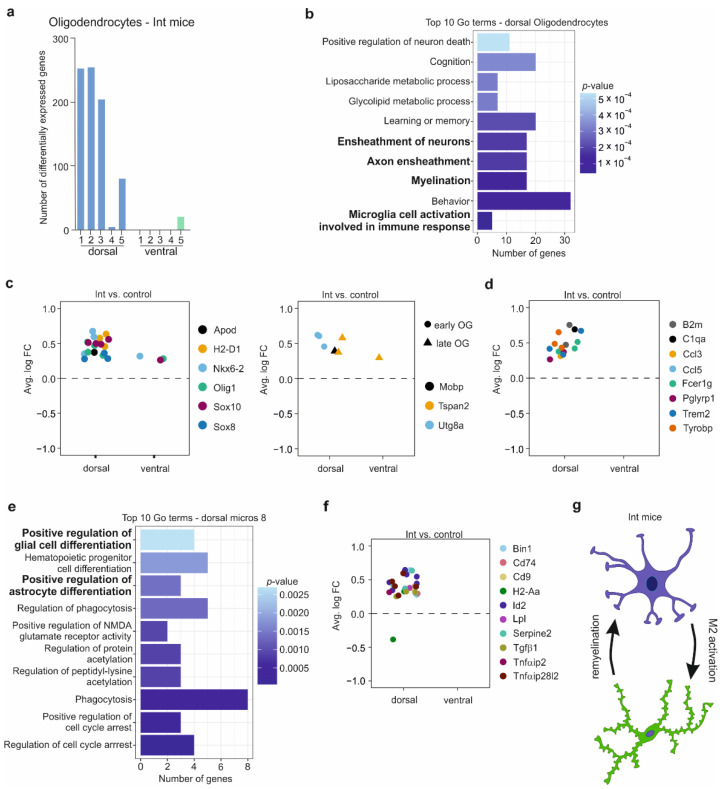
Chronic social defeat stress induces an oligodendrocyte response in the hippocampal dorsal part in the intermediate mice. (**a**) Number of DEGs among the oligodendrocytes sub-clusters 1 to 5 in the intermediate mice compared to control mice in the dorsal and ventral parts of the hippocampus. (**b**) Top 10 GO terms analysis related to the DEGs identified in the dorsal hippocampus of the intermediate mice. The most relevant pathways are highlighted in bold. (**c**) Differential expression of the oligodendrocyte progenitor (left panel), early and mature (right panel) oligodendrocyte markers in the intermediate (Int) mice compared to control mice in the dorsal and ventral parts of the hippocampus. (**d**) Differential expression of oligodendrocyte genes involved in microglia cell activation participating in the immune response pathway in the intermediate mice compared to control mice in the dorsal and ventral parts of the hippocampus. (**e**) Top 10 GO terms analysis related to the DEGs identified in the dorsal hippocampus of the sub-cluster micros_8 of the intermediate mice. The most interesting pathways are highlighted in bold. (**f**) Differential expression of the genes involved in the positive regulation of the glial cell differentiation pathway in the microglia cluster in the intermediate mice compared to control mice in the dorsal and ventral parts of the hippocampus. (**g**) Proposed cell-cell interaction identified in the intermediate mice between oligodendrocytes (top) and microglia (bottom).

**Figure 5 cells-11-03405-f005:**
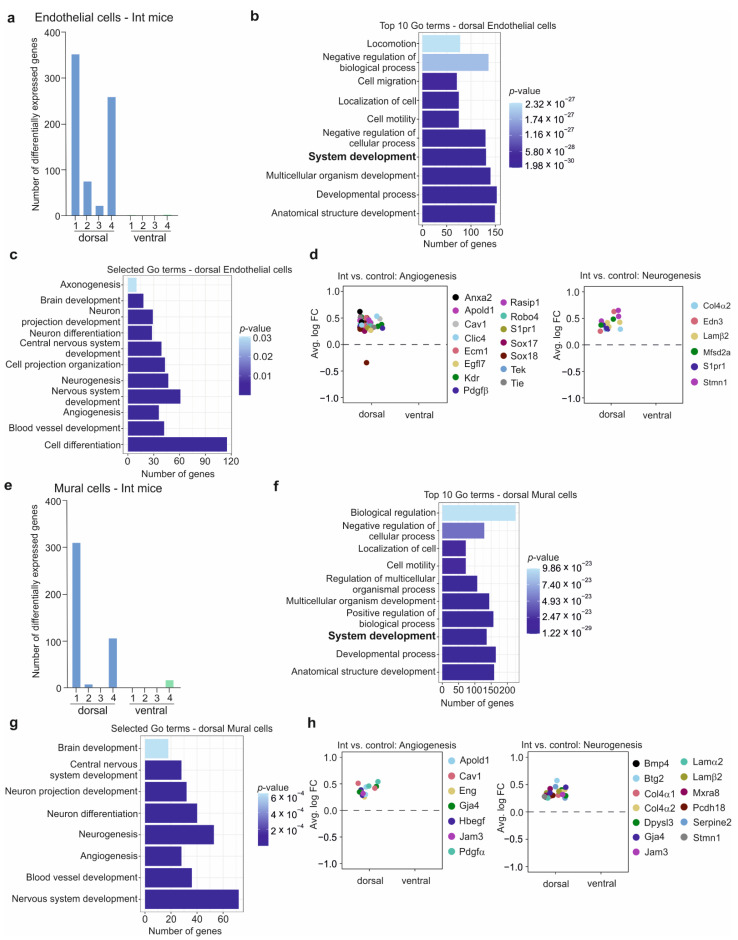
Chronic social defeat stress induces an endothelial and mural cells response in the hippocampal dorsal part in the intermediate mice. (**a**) Number of DEGs among the endothelial sub-clusters 1 to 4 in the intermediate mice compared to control mice in the dorsal and ventral parts of the hippocampus. (**b**) Top 10 GO terms analysis related to the DEGs identified in the endothelial cells in the dorsal hippocampus of the intermediate mice. The most interesting pathways are highlighted in bold. (**c**) Selected GO terms related to the DEGs identified in the endothelial cells in the intermediate mice’s dorsal hippocampus. (**d**) Differential expression of the genes involved in the angiogenesis (left panel) and neurogenesis pathways (right panel) in the endothelial cells of intermediate (Int) mice compared to control mice in the dorsal and ventral parts of the hippocampus. (**e**) Number of DEGs among the mural sub-clusters 1 to 4 in the intermediate mice compared to control mice in the dorsal and ventral part of the hippocampus. (**f**) Top 10 GO terms analysis related to the DEGs identified in the mural cells in the dorsal hippocampus of the intermediate mice. The most interesting pathways are highlighted in bold. (**g**) Selected GO terms related to the DEGs identified in the mural cells in the dorsal hippocampus of the intermediate mice. (**h**) Differential expression of the genes involved in the angiogenesis and neurogenesis pathways in the mural cells of intermediate mice compared to control mice in the dorsal and ventral parts of the hippocampus.

**Figure 6 cells-11-03405-f006:**
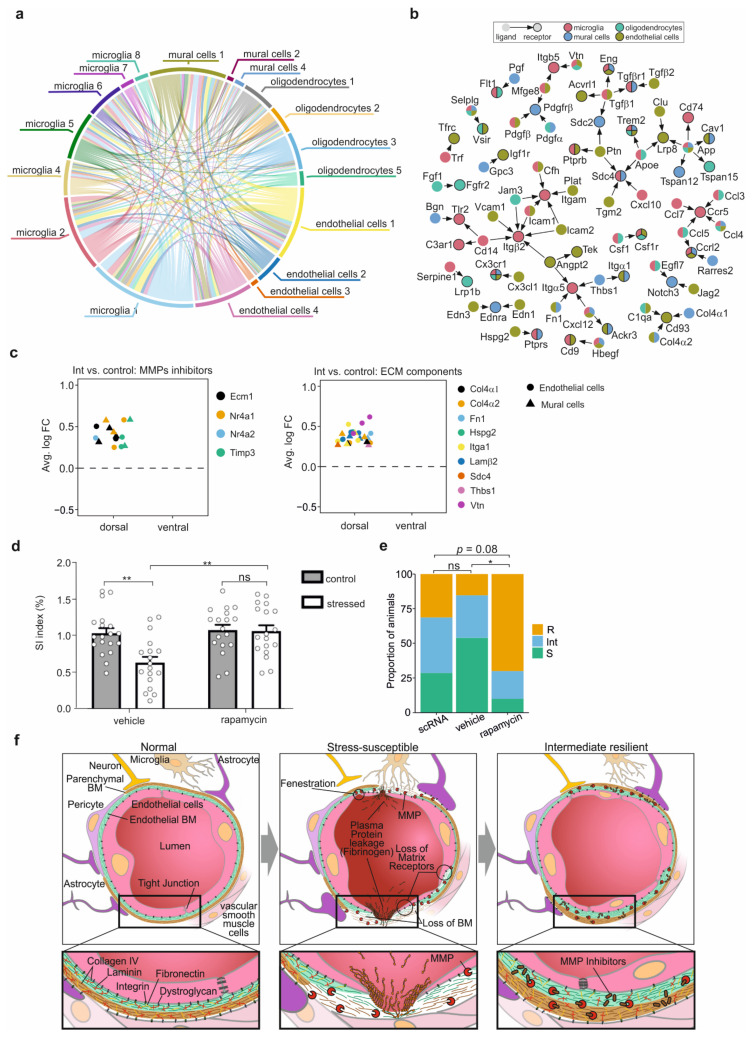
A complex stress response to chronic social defeat in intermediate mice identifies a potential pharmacological target promoting stress resilience. (**a**) Circos plot representing the putative ligand receptor interactions between the brain cell types and upregulated in the dorsal hippocampus of intermediate mice. The edges are coloured according to the ligand-generating cell type. The thickness is proportional to the number of connections identified. (**b**) Ligand-receptor interaction network identified in the dorsal hippocampus of intermediate mice. The arrows point out from the ligands to the receptors. The colour of nodes corresponds to the brain cell types expressing the ligands and/or the receptors. (**c**) Differential expression of the matrix metalloproteinases (MMPs) inhibitors (upper panel) and extracellular matrix (ECM) components (lower panel) in the endothelial and mural cells of intermediate (Int) mice compared to control mice in the dorsal and ventral parts of the hippocampus. (**d**) Social interaction (SI) index of controls and defeated mice treated with vehicle or rapamycin after CSDS during 7 days. Rapamycin treatment after CSDS increases SI index of defeated animals and promotes stress resilience. (** *p* < 0.01 Two-way ANOVA followed by Tukey’s post hoc test.) (**e**) Distribution of the proportion of animals assigned to the three behavioural phenotypes (R, Int and S) in the different behavioural experiments performed in the present work: for the single-cell RNA-seq (scRNA), and for the pharmacological intervention (vehicle and rapamycin) experiment. A significant increase of resilient mice was observed in the rapamycin treated group compared to the non-treated group. (* *p* < 0.05 Chi squared test) (**f**) Overview of the proposed molecular mechanism observed at the blood-brain barrier. In defeated animals, the endogenous expression of MMPs inhibitors or the rapamycin treatment could promote stress resilience by maintaining the ECM integrity.

## Data Availability

All data needed to evaluate the conclusions in the paper are present in the paper and/or the Appendix A. Additional data related to this paper maybe requested from the authors.

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
