# Peer review of "A Resilience Related Glial-Neurovascular Network Is Transcriptionally Activated after Chronic Social Defeat in Male Mice"

_cells, 2022, doi:10.3390/cells11213405_

Round 1
Reviewer 1 Report
The molecular mechanisms of resilience following chronic stress is of great importance and not fully understood. Using a well-established mouse model, chronic social defeat stress (CSDS), the authors profiled the hippocampal single cell transcriptome from mice displaying susceptible, intermediate and resilient behavior following stress and compared their transcriptome with the control mice. They have identified cell-type specific transcriptional response in the defeated mice, almost exclusively from the intermediate group. Finally, using an mTOR pathway inhibitor, rapamycin, they have demonstrated a shift in SI index from the susceptible group to the resilient group of mice treated with rapamycin. Overall, the study could serve as a useful resource as a cell-type specific transcriptome in the hippocampus following CSDS. A few clarifications below could help further improve the manuscript:
1. Additional technical details are required for the starting materials used in the study. Once the tissue were dissected and cells were sorted, the cells were frozen and thawed before loading onto the chips. This freeze-thaw step is generally not recommended and known to result in cell loss. What is the recovery rate from the freeze-thaw cycle? Are the cells still viable and healthy after this freeze-thaw cycle? In addition, 3500 cells were loaded on to the chip. Given the 10X chip is known to have a multiplet rate, which results in lower cell number recovered, surprisingly 4000 cells be captured from the bioinformatics analysis? What is the criteria used for cell capture analysis? How can 4000 cells be captured if Cellranger was run with force-cell 2000 setting?
2. Using the MSI test to further categorize the defeated animal is a new approach adopted by the authors’ group. The intermediate group represents a group of animals which were able to discriminate aversive from non-aversive stimulation, and is worth further investigation. In fact, they were almost the only group which presented a differential single cell transcriptome from the control group. How do the authors interpret these uniquely identified group, and the lack of identification of DEGs from the conventional susceptible and resilient mice, especially that there were plenty of bulk cell RNA-seq studies characterizing their transcriptomic difference from the same region (Bagot et al., Neuron, 2016)? Could these results be attributed to the unique testing approach of discrimination rather than susceptible or resilience?
3. The rapamycin treatment is a good rescue experiment, but seems not well unjustified or unrelated to the study. Even though the mTOR kinase is a crucial effector of many of the pathways identified here, were any of the mTOR signaling pathway regulators differentially expressed in any of the cell types? If so, manipulating the mTOR pathways in those given cell types should be performed specifically. If not, a global manipulation can be quite unspecific and the effect may well be secondary. In addition, it is unclear if the animals treated with rapamycin were subjected to SI test earlier and displayed susceptible phenotype. Also, given the most DEGs were identified from the intermediate group, shouldn’t the rapamycin-treated animals be tested with the MSI test, and analysis should be performed with the intermediate group specifically?
4. Since the fold change cutoff for DEG analysis is rather small (0.25), validation of top DEG should be performed.
5. Line 678-678, “We did not detect molecular changes which would indicate an active stress resilience process, but rather suggesting these animals displayed a passive resilience type”. Please clarify this sentence.
6. Line 683-684, “In contrast, in the S mice, only weak molecular changes were observed, suggesting that these animals were truly susceptible”. It was not clear what this sentence means.
Author Response
Reviewer 1
The molecular mechanisms of resilience following chronic stress is of great importance and not fully understood. Using a well-established mouse model, chronic social defeat stress (CSDS), the authors profiled the hippocampal single cell transcriptome from mice displaying susceptible, intermediate and resilient behavior following stress and compared their transcriptome with the control mice. They have identified cell-type specific transcriptional response in the defeated mice, almost exclusively from the intermediate group. Finally, using an mTOR pathway inhibitor, rapamycin, they have demonstrated a shift in SI index from the susceptible group to the resilient group of mice treated with rapamycin. Overall, the study could serve as a useful resource as a cell-type specific transcriptome in the hippocampus following CSDS. A few clarifications below could help further improve the manuscript:
Point 1.
Additional technical details are required for the starting materials used in the study. Once the tissue were dissected and cells were sorted, the cells were frozen and thawed before loading onto the chips. This freeze-thaw step is generally not recommended and known to result in cell loss. What is the recovery rate from the freeze-thaw cycle? Are the cells still viable and healthy after this freeze-thaw cycle?
Answer: After the freeze-thaw step, the recovery rate was around 7% of the total cell number with a cell viability of these recovered cell population at around 67%. Surprisingly, the cell viability measured before the 10X loading and the % of mitochondria reads detected in the same sample did not correlate. In each sample, the % of mitochondria reads was around 5% as shown in the Supplementary Figure 1e. According to the 10X recommendations, to isolate 2,000 single cell per sample, 3,500 cells were loaded.
In addition, 3500 cells were loaded on to the chip. Given the 10X chip is known to have a multiplet rate, which results in lower cell number recovered, surprisingly 4000 cells be captured from the bioinformatics analysis? What is the criteria used for cell capture analysis? How can 4000 cells be captured if Cellranger was run with force-cell 2000 setting?
Answer: Once each sample passed our filtering criteria (between 200 and 2,500 reads to exclude any cell free RNA or multiplets, <15% of mitochondrial gene reads to remove any dead cells), the reads of the left and right parts of the dorsal and ventral hippocampus were pooled together, leading to a maximum of 4,000 cells per each sample group for the downstream analysis.
For improved clarity, we edited the materials and methods part and added Table 1.
Point 2.
Using the MSI test to further categorize the defeated animal is a new approach adopted by the authors’ group. The intermediate group represents a group of animals which were able to discriminate aversive from non-aversive stimulation, and is worth further investigation. In fact, they were almost the only group which presented a differential single cell transcriptome from the control group. How do the authors interpret these uniquely identified group, and the lack of identification of DEGs from the conventional susceptible and resilient mice, especially that there were plenty of bulk cell RNA-seq studies characterizing their transcriptomic difference from the same region (Bagot et al., Neuron, 2016)?
Answer: In all the bulk transcriptomic analyses following CSDS, the resilient versus susceptible classification was performed 1 day after the last day of stress, and the tissues were collected on the following day, i.e., 2 days after the last day of stress. In our analysis, the animals were classified 5 and 8 days after the last day of stress, and the hippocampi were collected 10 days after the last day of stress. A longitudinal analysis at day 1 and day 10 after stress previously identified dynamic changes in resilient and in susceptible mice (Bouvier et al., 2017). Indeed, the authors showed a downregulation of BDNF in resilient and susceptible mice 1 day after the stress compared to the control mice. However, 10 days after the stress, BDNF was downregulated only in the susceptible mice compared to the control; no differences between the resilient and the control mice were observed anymore. This study highlighted a dynamic and active change in the resilient mice over the time of 10 days, suggesting the activation of recovery mechanisms.
In our analysis, we collected the samples 10 days after the last day of stress, whereas Bagot et al. (2016) collected the samples 2 days after the stress. The different time points of tissue collection, i.e., 2 days versus 10 days, likely explains why we could not detect any major transcriptional changes in the resilient animals compared to the control mice.
Could these results be attributed to the unique testing approach of discrimination rather than susceptible or resilience?
Answer: In all the transcriptomic analyses, no information regarding the SI index of the mice selected for the bulk analyses was provided. However, in our study, we addressed the significant heterogeneity among the classical susceptible mice, justifying the sub-classification we performed with the identification of the intermediate sub-group of mice. We observed major transcriptional changes only in the intermediate animals. Yet, we cannot exclude that the signatures observed in the classical heterogeneous group of susceptible mice from previous bulk transcriptomic analyses originated from the intermediate sub-group which drew our attention.
These thoughts are now included in the discussion.
Point 3.
The rapamycin treatment is a good rescue experiment, but seems not well justified or unrelated to the study. Even though the mTOR kinase is a crucial effector of many of the pathways identified here, were any of the mTOR signaling pathway regulators differentially expressed in any of the cell types?
Answer: The reviewer raises important points. The mTOR pathway activation leads to the phosphorylation of several proteins including eIF2alpha, a post-translational modification that cannot be detected in transcriptomic datasets but only at the protein level. This is the reason why we could not analyze it in our dataset. However, we looked at the last steps of mTOR pathways and found relevant genes to be altered, suggesting that the mTOR pathways are indeed involved in stress resilience (Figure 6c). Moreover, a KEGG pathway analysis revealed the upstream negative regulators of mTOR, the MAPK and PI3K-AKT signaling pathways, are significantly upregulated in the dorsal hippocampus of Int mice compared to control mice (new Supplementary Figure 9). In line with this, it has been already reported that mTOR inhibition with rapamycin induces MAPK and PI3K-Akt signaling pathway (Albert et al., 2009). Altogether, we were able to collect convincing evidence supporting our hypothesis that mTOR inhibitors, such as rapamycin, would activate stress resilience mechanisms in the susceptible mice.
If so, manipulating the mTOR pathways in those given cell types should be performed specifically. If not, a global manipulation can be quite unspecific and the effect may well be secondary.
Answer: The goal of the study was to identify a network of genes involved in stress resilience, which could be targeted by pharmacological intervention, also in humans. Therefore, for a better translational application, we decided to deliver rapamycin with systemic injection instead of brain region specific (i.e., dorsal hippocampus specific) injection. Moreover, rapamycin is a crucial effector of cerebrovascular functions, regulating neurogenesis, myelination, synaptic plasticity, microglia activation and polarization, and influences the permeability of the BBB and the composition of the ECM (to cite a few papers: Bockaert et al., 2015; You et al., 2016; Xu et al., 2021; Chen et al., 2016; Harry, 2013; Van Vliet et al., 2016; Chi et al., 2017; Van Skike et al., 2020). Furthermore, an overexpression of the MAPK and PI3K-Akt signaling pathways have been observed in the dorsal hippocampus of Int mice (new Supplementary Figure 9) and after mTOR inhibitor treatment (Albert et al., 2009). Therefore, we do not consider that the observed rapamycin-induced stress resilience-promoting effects are due to side effects.
In addition, it is unclear if the animals treated with rapamycin were subjected to SI test earlier and displayed susceptible phenotype.
Answer: Prior to the rapamycin treatment, the animals were not subjected to any SI or MSI test in order to avoid the re-testing effects. After the 10 days of stress, the animals were treated with either vehicle or rapamycin by intraperitoneal injection for the following 7 days. The SI test was performed after the treatment in drug-free conditions.
Also, given the most DEGs were identified from the intermediate group, shouldn’t the rapamycin-treated animals be tested with the MSI test, and analysis should be performed with the intermediate group specifically?
Answer: As said above, rapamycin is a crucial effector of cerebrovascular functions, regulating neurogenesis, myelination, synaptic plasticity, microglia activation and polarization and influencing the permeability of the BBB and the composition of the ECM. These effects were similar to the mechanisms we observed in the intermediate group of mice. Therefore, we hypothesized that rapamycin treatment after stress could either enhance or activate the mechanisms we observed in the intermediate group to the entire group of the susceptible animals (SI<0.75) and therefore is able to promote stress resilience. After rapamycin treatment, we indeed observed a significant increase of resilient animals (Figure 6d,e). To analyze the effect of the treatment on the stress resilience, the animals were reassigned to three stress resilience groups based on their SI index, i.e., R with SI index <1.15; S with SI index <0.5 and Int with SI index between 0.5 and 0.75, similarly to the Fig 1d. The proportion of animals in the intermediate group remained unaltered in both conditions (vehicle or treated), suggesting that this group is indeed an intermediate step towards stress resilience. The loss of the susceptible sub-group in the rapamycin-treated group revealed that rapamycin is able to promote stress resilience. Overall, the rapamycin treatment induced a longitudinal shift in the group-size from the susceptible to the resilient group.
Point 4.
Since the fold change cutoff for DEG analysis is rather small (0.25), validation of top DEG should be performed.
Answer: Due to the small fold change (FC) for some of the genes identified, it might be difficult to detect any significant differential expression with qPCR. Indeed, the differential expression detected in specific cell type will be diluted by the other cell types present in the sample. Moreover, even though we would have not been able to detect any significant changes by qPCR, we are convinced that one gene alone is not enough to activate the potential mechanisms we identified but rather their combination/network. Indeed, currently, network analyses are the state-of-the-art for transcriptomic or system analysis that have unraveled new networks for new therapeutic interventions. For example, organ system analysis in critically ill patients unraveled an unstable cardiovascular-hepatic-coagulation network in the non-survivor group (Asada et al., 2019, Critical care, Organ system network analysis and biological stability in critically ill patients). Similarly, multi-omics network analyses identified gene networks in Alzheimer disease patients and predicted the repurposing of existing FDA approved drugs for therapeutic intervention (Gupta et al., 2022, Plos Computational Biology, Single-cell network biology characterizes cell type gene regulation for drug repurposing and phenotype prediction in Alzheimer’s disease). In line with these studies, our goal was to identify networks of genes involved in stress resilience. We are convinced that the cumulative effects of the differential expression of each gene identified, even though the fold change (FC) is rather small, are sufficient to activate the stress resilience biological processes identified.
Point 5.
Line 678-678, “We did not detect molecular changes which would indicate an active stress resilience process, but rather suggesting these animals displayed a passive resilience type”. Please clarify this sentence.
Answer: For better clarity, the sentence was changed.
Point 6.
Line 683-684, “In contrast, in the S mice, only weak molecular changes were observed, suggesting that these animals were truly susceptible”. It was not clear what this sentence means.
Answer: The sentence was corrected.
Reviewer 2 Report
In this manuscript, Vennin et al. investigate the molecular mechanisms underlying stress resilience. Using the chronic social defeat stress (CSDS) model, the authors stratified defeated mice into resilient, susceptible-intermediate, and susceptible groups after 5 days to study long-term stress responses. The authors then used single cell RNA-sequencing to define molecular motifs that could drive behavioral heterogeneity in CSDS. Prominent gene expression signatures included disrupted glia and endothelial pathways, but only in mice defined as susceptible-intermediate and only in dorsal hippocampus. The approach is exciting and will likely address key issues in the field. However, in my view the data appears weak and does not substantiate the claim that perturbed glia-neurovascular networks promote resilience.
Main Points:
1.) In figure 1b-c, the authors define CSDS mice as resilient or susceptible based on social interaction scores of >1.15 and <0.75. How were these defined? The idea that susceptible mice display heterogenous responses is central to the manuscript and used to define susceptible-intermediate groups. However, this is entirely dependent on where the threshold for resilient vs susceptible is placed.
2.) In figure 1e, the authors show that susceptible intermediate mice in the modified social interaction test display increased preference for 129 mice as a “resilient” phenotype, but this does not occur in the resilient group. Can the authors talk about this?
Additionally, can the authors show that these intermediate-susceptible mice are truly intermediate in their chronic stress responses in orthogonal tests that assay chronic stress levels, such as open field or elevated plus maze tests?
3.) The authors used rapamycin to reverse the gene expression changes observed in susceptible-intermediate mice and find a rescue in CSDS. However, rapamycin treatment broadly affects many pathways throughout the body and therefore does not substantiate the claim that the pathways highlighted by the authors are driving stress susceptibility. Could the authors do localized injections of rapamycin in the dorsal vs ventral hippocampus to determine if the expected CSDS rescue occurs in dorsally injected mice vs ventrally injected? The authors also do not show that the proposed pathways are actually modulated by rapamycin in dorsal vs ventral hippocampus of intermediate-susceptible mice in vivo.
4.) In figure 2, it’s strange to me that there are virtually no transcriptomic differences in the hippocampus between control mice and CSDS (besides the susceptible-intermediate group). This is especially strange because the susceptible-intermediate group are not a discrete population- they exist in a continuum with the susceptible mice so I would expect at least a partial overlap in DEGs in the susceptible group. This issue is further highlighted in figure 1c, where the susceptible and susceptible-intermediate mice are indistinguishable with respect to PC1 – are these results something the authors expected? Can the authors comment more on these findings?
5.) In figure 2b-c, there is low signal to noise in cell type-specific marker expression. For example, microglia appear to express high levels of astrocyte genes. The combination of warm papain dissociation, FACS sorting, and then freeze-thawing samples prior to scRNA-seq raises the issue of sample integrity. Most likely the single cell nature of the data is compromised due to these harsh conditions and there is likely high levels of ambient RNA contamination in each GEM that later is called as a cell by CellRanger.
Since the authors used 10x’s CellRanger pipeline, can they provide the GEX barcode rank plot? Also in supplemental figure 1, the authors do not show what their QC cutoffs are. The authors need to show what cutoff values for UMIs, genes, %mitochondria are used and a table showing how many cells were analyzed per condition per mouse. Without these metrics, assessing quality control is not possible.
Minor Points:
1.) In figure 1c, the authors use PCA to select susceptible-intermediate mice. What parameters were used to define this group? They do not appear to be separate populations. Also, can the authors define what metrics are used to define PC1 vs PC2?
2.) Why are the control datapoints different between figure 1b and 1d? Based on the description of the experiment, I thought these should be the same animals.
3.) The authors may be aware, but neurons do not typically survive the warm papain dissociation process, which also introduces many technical artifacts. A cold, mechanical dissociation to isolate neuronal nuclei is recommended for downstream single cell transcriptomic approaches.
Round 2
Reviewer 2 Report
The authors have addressed my concerns except for major point 3 (see below).
If they soften their claims and re-frame the paper as a resource paper it will be acceptable for publication
Major Point 3 response:
I agree that the systemic Rapamycin approach is in line with potential translational applications and I do not think systematically testing every pathway identified is necessary.
However, the issue I have is that the manuscript claims that “A glial-neurovascular network promotes resilience after chronic social defeat” in the title and throughout the manuscript. There is no direct evidence of this.
The evidence presented here shows that there are broad glia/vascular-related transcriptional changes that occur after stress and this coincides with a behavioral “rescue” with rapamycin treatment. Rapamycin is known to affect many of these pathways, but this does not mean that these glial-neurovascular networks promote the behavioral phenotype. Rapamycin could be doing something completely different to mediate the behavioral effects and the authors would not know because they don’t directly test this, yet the central claim is that a glial-neurovascular network promotes resilience.
The suggested experiments were not intended to validate rapamycin’s well-studied effects – they are to strengthen the conclusions that the authors draw. Are the dramatic and surprising dorsal hippocampal changes that are detected really responsible for the behavioral phenotype? If many different pathways are affected by stress as detected by transcriptomics, and rapamycin has all of these multimodal effects as the authors have cited, how do you know that it is glial-neurovascular networks specifically that promote resilience? If the authors are unable to provide direct evidence for their claims, they need to soften their conclusions in the manuscript. Perhaps they can re-frame it as a resource paper as they have suggested in reviewer rebuttal comments.
Author Response
Reviewer 2
The authors have addressed my concerns except for major point 3 (see below).
If they soften their claims and re-frame the paper as a resource paper it will be acceptable for publication
Major Point 3 response:
I agree that the systemic Rapamycin approach is in line with potential translational applications and I do not think systematically testing every pathway identified is necessary.
However, the issue I have is that the manuscript claims that “A glial-neurovascular network promotes resilience after chronic social defeat” in the title and throughout the manuscript. There is no direct evidence of this.
The evidence presented here shows that there are broad glia/vascular-related transcriptional changes that occur after stress and this coincides with a behavioral “rescue” with rapamycin treatment. Rapamycin is known to affect many of these pathways, but this does not mean that these glial-neurovascular networks promote the behavioral phenotype. Rapamycin could be doing something completely different to mediate the behavioral effects and the authors would not know because they don’t directly test this, yet the central claim is that a glial-neurovascular network promotes resilience.
The suggested experiments were not intended to validate rapamycin’s well-studied effects – they are to strengthen the conclusions that the authors draw. Are the dramatic and surprising dorsal hippocampal changes that are detected really responsible for the behavioral phenotype? If many different pathways are affected by stress as detected by transcriptomics, and rapamycin has all of these multimodal effects as the authors have cited, how do you know that it is glial-neurovascular networks specifically that promote resilience? If the authors are unable to provide direct evidence for their claims, they need to soften their conclusions in the manuscript. Perhaps they can re-frame it as a resource paper as they have suggested in reviewer rebuttal comments.
Answer: Indeed, no direct evidences between the rapamycin treatment and the molecular response we identified were provided in the manuscript. Therefore, the manuscript has been edited according to your suggestions (see in Abstract, Introduction, Result, and Discussion). We explicitly specified that the manuscript should be consider as a resource paper. Accordingly, we also changed the title to: “A resilience related glial-neurovascular network is transcriptionally activated after chronic social defeat in male mice”.